# The Role of microRNAs in Inflammation

**DOI:** 10.3390/ijms232415479

**Published:** 2022-12-07

**Authors:** Kaushik Das, L. Vijaya Mohan Rao

**Affiliations:** Department of Cellular and Molecular Biology, The University of Texas Health Science Center at Tyler, Tyler, TX 75708, USA

**Keywords:** inflammation, microRNA, small non-coding RNA

## Abstract

Inflammation is a biological response of the immune system to various insults, such as pathogens, toxic compounds, damaged cells, and radiation. The complex network of pro- and anti-inflammatory factors and their direction towards inflammation often leads to the development and progression of various inflammation-associated diseases. The role of small non-coding RNAs (small ncRNAs) in inflammation has gained much attention in the past two decades for their regulation of inflammatory gene expression at multiple levels and their potential to serve as biomarkers and therapeutic targets in various diseases. One group of small ncRNAs, microRNAs (miRNAs), has become a key regulator in various inflammatory disease conditions. Their fine-tuning of target gene regulation often turns out to be an important factor in controlling aberrant inflammatory reactions in the system. This review summarizes the biogenesis of miRNA and the mechanisms of miRNA-mediated gene regulation. The review also briefly discusses various pro- and anti-inflammatory miRNAs, their targets and functions, and provides a detailed discussion on the role of miR-10a in inflammation.

## 1. Introduction

Inflammation is the immune system’s response against harmful stimuli, such as pathogens [1], toxic compounds [2], damaged cells [3], or radiation [4], that aids in the removal of injurious stimuli and starts the healing process in the damaged tissue [5]. Inflammation thus acts as a defense mechanism of the system against harmful stimuli [6]. However, uncontrolled inflammation often leads to various chronic inflammatory syndromes [7]. Therefore, a balance between the pro- and anti-inflammatory signals in the host immune system is crucial to clearing the pathogen without causing extensive damage to the host. Inflammation causes redness, tissue swelling, pain, heat generation, and a loss of tissue functions. It often results in increased vascular permeability and the recruitment of leukocytes to the infection site [8,9]. Recruited leucocytes are activated at the injury site and release cytokines, which further extend the inflammatory response [10].

In the past few decades, one of the most important transformations in RNA biology is the discovery of small (~20–30 nucleotides) non-coding RNAs (small ncRNAs), which regulate gene expression at multiple levels. These regulations include the alteration in chromatin structures [11], segregation of chromosomes [12], transcriptional regulation [13], stability of RNA [14], and regulation of protein synthesis [15]. The regulatory effects of small ncRNAs are generally considered inhibitory, as in most instances, they inhibit the expression of their target genes. Based on their biological roles, origins, structures, and targeted effector molecules, small ncRNAs are classified into three distinct populations: microRNAs (miRNAs), short interfering RNAs (siRNAs), and Piwi-interacting RNAs (piRNAs). The differences among the three forms of small ncRNAs are very subtle. mi- and siRNAs are principally involved in the regulation of gene expression in eukaryotes [16]; however, in a few instances, they have also been found to play important roles in regulating gene expression in prokaryotes [17,18]. In contrast, the piRNAs are found to be actively involved in gene regulation in both eukaryotes and prokaryotes [19,20]. The precursor molecules for mi- and siRNAs are known to be double-stranded [14], whereas piRNAs’ precursors are believed to be single-stranded [21]. The effect of miRNAs is observed in both the germlines and somatic lines of eukaryotes [22], whereas piRNAs exert their functions more specifically in the germlines of eukaryotes [23]. Moreover, mi- and siRNAs bind to the effector proteins, which belong to the AGO clade of Argonaute proteins, whereas piRNAs bind to the Piwi clade proteins [14]. 

In the present review, we will focus on how the distinct classes of miRNAs influence inflammatory responses in eukaryotes. We first discuss the biogenesis of miRNAs, the mechanism of their action at the cellular level, and the regulation of their expression in eukaryotic systems. Next, we briefly describe various pro- and anti-inflammatory miRNAs and how they influence inflammation. Lastly, we provide a detailed discussion on miR-10a, one of the prevalent anti-inflammatory miRNAs, on its chromosomal location, regulation of its expression, its role in various inflammation-associated diseases, and its atypical functions in the cells.

## 2. Biogenesis of miRNAs

The biogenesis of miRNAs starts with the co- or post-transcriptionally processing of the transcripts synthesized by RNA polymerase II/III [24]. Half of the presently identified miRNAs are derived from the processing of introns or a few protein-coding genes and are classified as intragenic. In contrast, other miRNAs are intergenic, i.e., independently transcribed and regulated through their own promoters [25,26]. Often, the transcription of miRNAs occurs as a long transcript designated as ‘clusters’ with similar seeding regions and thereby known as the miRNA family [27]. The biogenesis of miRNAs is comprised two basic pathways: canonical and non-canonical.

In the canonical biogenesis pathway (Figure 1, left panel), the dominant pathway of miRNA biogenesis, the pri-miRNAs, transcribed from their own genes, are processed to generate the pre-miRNAs by a protein complex consisting of DiGeorge Syndrome Critical Region 8 (DGCR8) and an RNase III enzyme family protein called Drosha [28]. pri-miRNA’s N6-methyladenylated GGAC and other motifs are recognized by DGCR8 [29], followed by subsequent cleavage of the pri-miRNA duplex at the base of the characteristic hairpin structure by Drosha to generate 2–3 nucleotides 3’ overhang on pre-miRNAs [30]. The pre-miRNAs thus generated are readily exported to the cytoplasm by the exportin-5/RanGTP complex and subsequently processed by the RNase III endonuclease, Dicer [28,31]. Dicer removes the terminal loop of pre-miRNAs to produce a mature miRNA duplex [32]. The 5’ end of the pre-miRNAs’ hairpin generates the 5p strand, whereas the 3p strand originates from the 3’ end of the pre-miRNAs after Dicer cleavage. Then, an ATP-driven process loads both strands of the mature miRNA duplex into the Argonaute (AGO) family proteins (AGO1-4 in humans) [33]. The stability at the 5’ end of the miRNA duplex, or 5’ U at nucleotide position 1, further determines the selection of 5p or 3p [34]. The AGO preferentially picks up the strand with lower 5’ stability, or 5’ U. The other strand, the passenger strand, is subsequently cleaved by AGO2 if no mismatch is found; otherwise, the miRNA duplex with a central mismatch is passively degraded [24].

The non-canonical pathway of miRNA biogenesis (Figure 1, right panel) consists of multiple pathways that involve different combinations of canonical pathway proteins, Drosha, Dicer, exportin-5, and AGO2. These pathways can be classified into two major groups, namely the DGCR8/Drosha-independent and the Dicer-independent pathways. The pre-miRNAs of one of the DGCR8/Drosha-independent pathways include mirtrons, which are generated from introns during mRNA splicing (Figure 1, right panel, ②) [35,36]. Another form of the DGCR8/Drosha-independent pathway is the 7-methylguanosine (m^7^G)-capped pre-miRNAs (Figure 1, right panel, ③), which are released into the cytoplasm via exportin-1 and do not require the Drosha cleavage [37]. On the other hand, in the Dicer-independent pathway, miRNAs that include short hairpin RNA (shRNA) transcripts are further processed by Drosha (Figure 1, right panel, ①) and require AGO2 for their maturation in the cytoplasm [38]. pre-miRNA loading into AGO2 results in the slicing of the 3p strand via 3’-5’ trimming [39], followed by the maturation of the 5p strand. 

## 3. Mechanisms of miRNAs-Dependent Gene Regulation

miRNAs are usually known to bind to specific sequences at the 3’-UTR of their target mRNA molecules, resulting in either translational repression or the degradation of the target mRNA molecules [40,41]. miRNAs also bind to the 5’-UTR, the coding region, or the promoter region of mRNAs [42]. The binding of miRNA to either the 5’-UTR or the coding region of the target molecules results in the inhibition of gene expression [43,44]. In contrast, their binding at the promoter site enhances gene expression [45]. Here, we will discuss the different modes of eukaryotic gene regulation by miRNAs.

## 4. miRNA-Dependent Gene Silencing Requires the miRISC

The miRNA-induced silencing complex (miRISC) includes the miRNA guide strand and the AGO protein [46]. The complementary sequences in the target mRNAs (miRNA-response elements; MREs) define the target specificity of the miRISC. Depending on this complementarity, the target mRNA molecules are either sliced by AGO2 or degraded by the miRISC, resulting in translational inhibition [47]. A 100% complementation between the miRNA and mRNA would induce AGO2 endonuclease activity, which chops the target mRNA [47]. Interestingly, the extensive base-pairing also destabilizes the miRNA itself and thus decrease amounts of miRNA [48,49]. In most cases, incomplete complementation [50] results in the loss of AGO2’s endonuclease activity, thereby acting as a mediator of RNA interference (as do other family members of AGO in humans: AGO1, -3, and -4). Sometimes, a functional miRNA–mRNA interaction also occurs between nucleotides 2 to 7 at the 5’ seed region [42,51]. Moreover, additional base pairing at the 3’ end not only enhances the stability but also increases the specificity of the miRNA–target interaction [52]. Once the stable interaction is formed between miRNA and mRNA [50], the miRISC complex further recruits the scaffolding protein, GW182 [53], which recruits the effector molecules such as the poly(A)-deadenylase complexes PAN2-PAN3 and CCR4-NOT [54]. PAN2/3 initiates the poly(A)-deadenylation of the target mRNA molecule, which is completed by CCR4-NOT. The deadenylation process is accelerated due to the interaction between the tryptophan (W)-repeats of GW182 and poly(A)-binding protein C (PABPC) (Figure 2A, ①) [50]. This is followed by a decapping reaction at the 5’ end, mediated by the decapping protein 2 (DCP2) alongside other associators [54], and subsequent 5’-3’ degradation by XRN1 (Figure 2A, ②) [55].

## 5. miRNA-Induced Translational Activation

Apart from their common role in suppressing gene expression, miRNAs, in some instances, are also known to play an important part in the up-regulation of gene expression. For example, AGO2 and another microRNA-protein complex (microRNP) family protein, fragile-x-mental retardation-related protein 1 (FXR1), are shown to bind at the 3’-UTR AU-rich elements (AREs) and activate the translational process (Figure 2B, ①) [56]. Several miRNAs play a dual role in different cell cycle stages. For example, let-7 during cell cycle arrest is shown to activate the AGO2-FXR1-dependent translational process, whereas, in proliferating stages, it inhibits protein synthesis (Figure 2B, ②) [56]. The miRNA-mediated translational activation is also observed in quiescent cells, such as oocytes, which essentially require AGO2-FXR1 (Figure 2B, ①) [57,58]. Under specific conditions, such as amino acid deprivation, certain miRNAs are known to bind to the 5’-UTR of several mRNAs encoding ribonuclear proteins (RNPs), thereby aiding in their translational activation [59]. 

## 6. miRNAs and Inflammation

Inflammation is primarily regulated by miRNAs through their altered expression in certain immune cells [60]. As a part of the inflammatory response, the biogenesis of miRNAs is often regulated at different stages, such as the synthesis, processing, and stabilization of pre- or mature miRNAs [61,62]. miRNAs regulate different stages of inflammation, starting from initiation, expansion, and resolution by both positive and negative feedback [63]. In the positive feedback, the array of events restricts not only the invasion of pathogens but also the successful repair of tissue damage. In contrast, the negative feedback, activated during severe inflammation, helps maintain tissue homeostasis. In the following section, we briefly discuss how various pro- and anti-inflammatory miRNAs (Figure 3) exert their effects (Table 1 and Table 2). We limit our discussion to a few selective and prevalent miRNAs, with specific emphasis on miR-10a, one of the most abundant and prevalent endothelial anti-inflammatory miRNAs associated with several disease conditions. 

## 7. Pro-Inflammatory miRNAs

miR-155 is considered one of the most abundant pro-inflammatory miRNAs [88], expressed in a wide variety of cells, such as monocytes, macrophages, activated B cells, T cells, etc., and allows the translation of pro-inflammatory cytokine, TNF-α [64]. The expression of miR-155 is often induced by LPS [64], and the anti-inflammatory cytokine IL-10 is known to down-regulate miR-155 expression [89]. LPS governs the expression of miR-155 via the activation of the MyD88 and TRIF pathways [88]. miR-155 knock-out mice are shown to have an impaired immune response against *Salmonella* infection due to defects in B- and T-cell activation and are difficult to immunize against this pathogen [90]. Another study indicated that miR-155 knock-out mice show a significant reduction in the number of B-cell germinal centers, whereas miR-155 over-expressive mice show elevated numbers of them [91]. All these studies indicate the importance of miR-155 in the pro-inflammatory response. 

The expression of another pro-inflammatory miRNA, miR-92a, is significantly up-regulated in atherogenic endothelial cells (EC), and the transfer of miR-92a via extracellular vesicles (EVs) from EC to macrophages results in the up-regulation of several pro-inflammatory genes in the recipient macrophages [65]. miR-200 family miRNAs show pro-inflammatory response via targeting Zeb-1 and up-regulating cyclooxygenase-2 and MCP-1 in vascular smooth muscle cells in the type 2 diabetic murine model [66]. Another miRNA, miR-23a, not only activates pro-inflammatory NF-ĸB signaling via targeting A20, but also suppresses the anti-inflammatory JAK1/STAT6 pathway upon targeting both JAK1 and STAT6 directly in macrophages [67]. miR-27a shows the same phenotypic response upon targeting IRF4 and PPAR-γ [67]. 

miR-29c is shown to be involved in exerting a pro-inflammatory response in patients with diabetic nephropathy by targeting tristetraprolin (TTP) [68]. miR-138 participates in the macrophage inflammatory response via targeting SIRT1 and activating the NF-ĸB signaling pathway [69]. miR-34 family miRNAs, such as miR-34a and -34c, are also shown to induce the release of pro-inflammatory cytokines and chemokines in the wound-edge epidermal keratinocytes of venous ulcers via targeting LGR4, thereby delaying the wound closure and contributing the pathological roles in venous ulcers [70]. 

Another miRNA, miR-132, is found to be up-regulated in LPS-challenged THP-1 macrophages [92] and associated with the pro-inflammatory response via the release of IL-8 and MCP-1 by regulating SIRT1 in starved adipose-derived stem cells [71]. miR-132 could be a biomarker for inflammatory bowel disease (IBD) [93] and rheumatoid arthritis (RA) [94]. Let-7a confers a pro-inflammatory response via targeting IĸBβ, leading to NF-ĸB activation and subsequent expression of inflammatory and adhesion molecules in endothelial cells [72].

## 8. Anti-Inflammatory miRNAs

miR-7 and miR-10a are the most abundant anti-inflammatory miRNAs and are known to be associated with various disease conditions. We will discuss them in detail, particularly miR-10a, after a brief discussion of other anti-inflammatory miRNAs. 

Other than miR-10a and miR-7, several anti-inflammatory miRNAs are associated with various disease conditions. miR-126, another abundant endothelial miRNA, inhibits vascular inflammation by targeting the Sprouty-related EVH1 domain-containing protein 1 (SPRED1), phosphoinositol-3 kinase regulatory subunit 2 (PIK3R2), and vascular cell adhesion molecule 1 (VCAM1) in spinal cord injury (SCI), and miR-126 therapy could be used as a potential therapeutic approach in recovery after contusion in SCI [73]. 

Another anti-inflammatory miRNA, miR-146a, plays a pivotal role in the pathogenesis of diabetic nephropathy, and miR-146a deficiency leads to the increased expression of inflammatory cytokines IL-1β, IL-18, and other markers of inflammasome activation in macrophages [74]. miR-124 is also known to mediate cholinergic anti-inflammatory response by targeting STAT3 and TACE, thereby limiting IL-6 and TNF-α secretion from macrophages [75]. miR-125b confers its anti-inflammatory potential by targeting TRAF6-mediated MAPK and NF-ĸB signaling, thus regulating IL-1β-induced inflammatory gene expression in human osteoarthritic chondrocytes [76]. 

miR-31 reduces the inflammatory response and promotes the regeneration of colon epithelium in mice [77]. miR-210, on the other hand, prohibits the NF-ĸB inflammatory signaling upon targeting DR6 in osteoarthritis [78]. miR-24 limits aortic vascular inflammation by inhibiting chitinase 3-like 1 (Chi3l1)-induced synthesis of pro-inflammatory cytokines in macrophages and restricts abdominal aneurysm development in mice [79]. 

miR-149 shows its anti-inflammatory response via targeting TAK1/NF-ĸB signaling in osteoarthritic chondrocytes [80]. miR-181a down-regulates TNF-α, thereby improving renal inflammation in the diabetic nephropathy animal model [81]. The down-regulation of miR-150 induces the LPS-mediated release of pro-inflammatory cytokines in THP-1 macrophages via STAT1 up-regulation [82]. miR-143, on the other hand, down-regulates the TLR4/MyD88/NF-ĸB pathway and confers its anti-inflammatory potential in pulmonary epithelial cells [83]. miR-9 is known to inhibit the formation of inflammasomes and down-regulate inflammation in atherosclerosis by targeting the JAK1/STAT1 pathway in macrophages [84]. miR-142 also shows its anti-inflammatory response in murine macrophages through targeting IRAK1 and inhibiting the synthesis of pro-inflammatory NF-κB1, TNF-α, and IL-6 [85]. miR-223 modulates the inflammatory response by inhibiting IKKα and MKP5 in human gingival fibroblasts [86]. It also suppresses TLR4 signaling in macrophages [95] and intestinal inflammasome formation [96]. miR-21 plays a pivotal role in controlling excessive inflammation by DAMPs after MI via targeting KBTBD7 in macrophages, thereby making miR-21 a potential therapeutic target in the treatment of MI [87]. 

## 9. miR-7

miR-7 is animportant anti-inflammatory miRNA [97] thatplays significant roles in various diseases, such as cancer, cardiovascular diseases, and pregnancy-associated diseases (Table 3).

**Cancer:** miR-7 plays a pivotal role in the growth and development of various tumors. The expression of miR-7 is shown to be down-regulated in metastatic breast cancer tissues as compared to normal breast tissues, and the down-regulation of miR-7 often leads to the induction of breast cancer cell growth, invasiveness, migration, proliferation, and stemness, while preventing apoptosis [98]. Reddy et al. showed that miR-7 introduction into invasive breast cancer cells leads to the inhibition of cancer cell migration, invasiveness, and anchorage-independent growth via down-regulating its target, P21-activated kinase 1 (PAK1) expression [99]. Kong et al. [100] demonstrated that in breast cancer cells, miR-7 inhibits the expression of focal adhesion kinase (FAK) by directly targeting it. This leads to the suppression of epithelial–mesenchymal transition (EMT) and breast cancer metastasis [100]. miR-7 also targets Krüppel-like factor 4 (KLF4) in breast cancer stem-like cells (CSC) and impedes their metastasis to the brain [101]. Li et al. demonstrated that miR-7-dependent targeting of HoxB3 prevents tumor growth and suppresses the colony-formation ability of breast cancer cells [102]. Other studies showed that miR-7 perturbs the invasive potential of human breast cancer cells and sensitizes them to DNA damage via directly targeting SET domain-containing (lysine methyltransferase) 8 (SET8) [103].

In lung cancer, miR-7 down-regulation often leads to the progression of the tumor growth. For example, miR-7 suppresses the proliferation of lung cancer cells by targeting the epidermal growth factor receptor (EGFR) [104]. miR-7 is also shown to target the anti-apoptotic protein BCL-2 in lung cancer cells, which leads to the suppression of cell proliferation and the promotion of tumor cell apoptosis [105]. miR-7 is shown to downregulate the expression of proteasome activator 28 (PA28) subunit γ in non-small cell lung cancer, leading to the induction of apoptosis and cell cycle arrest [106]. miR-7 is also shown to target phosphoinositide-3-kinase regulatory subunit 3 (PIK3R3) in lung cancer cells, and the down-regulation of the PIK3R3/Akt pathway attenuates the TLR9 signaling-induced growth and proliferation of lung cancer cells [107].

A growing body of evidence indicates that miR-7 plays an important role in brain tumor development [108]. miR-7 is shown to inhibit glioblastoma progression by directly targeting EGFR and its downstream signaling molecules, PI3K and Raf-1 [109]. Similar to breast cancer, miR-7 also targets FAK1 in brain cancer cells, thereby inhibiting tumor cell proliferation and angiogenesis [110].

In colorectal cancer, miR-7 down-regulates the expression of paired box 6 (PAX6), which limits the PI3K/ERK-dependent up-regulation of MMP2 and MMP9 and hence inhibits colorectal cancer cell growth, proliferation, and metastasis [111]. miR-7 also inhibits the growth and metastasis of hepatocellular carcinoma by targeting AKT and down-regulating the PI3K/AKT pathway [112]. Zhao et al. showed that miR-7 inhibits gastric cancer metastasis by targeting insulin-like growth factor 1 (IGF-1) and IGF-1-mediated induction of EMT [113].

**Cardiovascular diseases:** Kaneto et al. demonstrated that miR-7 expression is up-regulated in the serum of patients suffering from left ventricular hypertrophy (LVH), and miR-7 serves as a biomarker for cardiovascular anomalies [114]. miR-7 is shown to suppress the growth and development of platelet-derived growth factor (PDGF)-BB-stimulated vascular smooth muscle cells (VSMCs) via targeting EGFR and could be used as a therapeutic regimen against cardiovascular diseases [115]. miR-7 is thought to serve as a biomarker for coronary atherosclerotic heart disease (CHD) [116] and cardiac sarcoidosis (CS) [117].

**Pregnancy-associated disease:** miR-7 was identified in the islet cells of the embryonic pancreas and could serve as a biomarker for diabetic embryopathy (DE) [118]. Complications during pregnancy often result from abnormal trophoblast invasion, and EMT plays a pivotal role in this process. miR-7 is shown to target EMT-related transcription factors and downregulate the mesenchymal markers, ultimately inhibiting trophoblast mesenchymal transition [119,120]. 

**Table 3 ijms-23-15479-t003:** The role of miR-7 in various diseases.

Disease	Cell Type	Expression	Target(s)	Function	Reference(s)
Cancer	BC cells	Down	PAK1	Inhibits cell migration, invasiveness, anchorage-dependent growth	[99]
	BC cells	Down	FAK	Inhibits BC metastasis	[100]
	BC stem-like cells	Down	KLF-4	Inhibits metastasis to the brain	[101]
	BC cells	Down	HoxB3	Inhibits tumor growth and colony-forming ability of BC cells	[102]
	BC cells	Down	SET8	Decreases invasive potential and sensitizes the cells to DNA damage	[103]
	LC cells	Down	EGFR	Suppresses proliferation	[104]
	LC cells	Down	BCL-2	Perturbs proliferation and promotes apoptosis	[105]
	Non-small LC cells	Down	PA28 subunit γ	Induction of apoptosis and cell-cycle arrest	[106]
	LC cells	Down	PIK3R3	Retards growth and proliferation	[107]
	Glioblastoma	Down	EGFR, PI3K, Raf-1	Prevents growth and development	[109]
	Brain cancer cells	Down	FAK1	Inhibits tumor proliferation and angiogenesis	[110]
	Colorectal cancer cells	Down	PAX6	Prevents growth, proliferation, and metastasis	[111]
	HC cells	Down	AKT	Inhibits growth and metastasis	[112]
	GC cells	Down	IGF-1	Inhibits GC growth	[113]
CVD	VSMCs	Up	EGFR	Suppresses VSMCs growth and development	[114]
DE	Islet cells	Up	EMT-TFs	Inhibits trophoblast mesenchymal transition	[119]

Abbreviations: BC, breast cancer; LC, lung cancer; HC, hepatocellular carcinoma; GC, gastric cancer; CVD, cardiovascular disease; VSMCs, vascular smooth muscle cells, DE, diabetic embryopathy.

## 10. miR-10a

miR-10a is considered to be a key post-transcriptional mediator in controlling inflammatory responses [121]. The actions of miR-10a are well-conserved among vertebrates, and its role is well-established in several inflammatory disorders such as rheumatoid arthritis (RA), inflammatory bowel disease (IBD), colitis, acute pancreatitis (AP), atherosclerosis, sepsis, cancer, etc. [122,123,124,125,126,127,128,129].

## 11. miR-10a Chromosomal Location

Genes encoding miR-10 family members are located within the homeobox (Hox) gene clusters [130], the transcription factors that critically regulate anterior–posterior patterning in bilaterian animals [131]. miR-10 is known to be co-expressed with Hox genes during development [132] and targets Hox transcripts [133], thereby believed to play a crucial role in determining body plans. Mammalian miR-10 family members miR-10a and miR-10b lie upstream of HoxB4 and HoxD4, respectively [134]. Due to the high degree of sequence conservation, differing only by a single nucleotide at the eleventh position (U and A for miR-10a and miR-10b, respectively), they target overlapping sequences in mRNAs [135]. Although miR-10a/b is shown to target Hox transcripts, a growing body of evidence indicates that various other pathways are also regulated by miR-10a/b [130]. Most of the mature miR-10 family members are generated from the 5’-arm of the hairpin precursors, whereas, in some instances, arm-switching results in the formation of mature miR-10 from the opposite arm [136].

## 12. Regulation of miR-10a Expression

Being a part of the Hox gene clusters, miR-10a/b are assumed to be regulated by cis-regulatory elements of the neighboring Hox genes. For example, prenatal exposure of fetal mouse brain to ethanol often leads to the co-expression of miR-10a/b and their associated Hox genes [137]. Moreover, miR-10a expression is often regulated by transcription factors p65 and TWIST1 (Figure 4) [138,139]. The inhibition of p65 nuclear translocation significantly reduces retinoic acid-induced miR-10a expression in embryonic stem cells, which prohibits their differentiation into smooth muscle cells [139]. On the other hand, TWIST1 is shown to enhance the expression of miR-10a in CD34^+^ cells in myelodysplastic syndrome, and TWIST1/miR-10a-axis could be used as a therapeutic target in the treatment of the myelodysplastic syndrome [138]. The regulation of miR-10a is also mediated by DNA methylation at the promoter region by knocking down both DNA methyltransferases, DNMT1 and DNMT3b, significantly increasing the expression of miR-10a in human colon cancer cells [140]. 

## 13. Targets of miR-10a in the Context of Various Inflammation-Associated Diseases

**Rheumatoid arthritis (RA):** miR-10a, a central regulator in the NF-ĸB signaling pathway, is often shown to regulate multiple inflammation-associated diseases (Figure 5 and Table 4). In RA patients, fibroblast-like synoviocytes (FLSs) are known to play an important role in cartilage and joint damage, deformation, and destruction [141]. Persistent activation of the NF-ĸB signaling pathway and the concomitant release of pro-inflammatory cytokines by the FLSs often contribute to the etiopathogenesis of RA [142]. Mu et al. observed a significant down-regulation of miR-10a expression in FLSs of RA patients as compared to osteoarthritis controls, leading to the activation of the NF-κB signaling pathway by up-regulating miR-10a target genes—interleukin-1 receptor-associated kinase 4 (IRAK4), transforming growth factor beta (TGF-β)-activated kinase 1 (TAK1). beta-transducin repeat containing E3 ubiquitin ligase (β-TrCP), and mitogen-activated protein 3 kinase 7 (MAP3K7) [126]. This leads induction of several pro-inflammatory cytokines, such as TNF-α, IL-1β, etc., which in turn down-regulate miR-10a expression via NF-κB-dependent activation of the transcription factor YY1 [126]. Thus, miR-10a plays a crucial role in controlling this regulatory circuit and could be an important therapeutic target for the treatment of RA. Hussain et al. have shown that in inflamed synoviocytes, the down-regulation of miR-10a promotes cell proliferation while restricting apoptosis, thereby contributing to the pathogenesis of RA [143]. Stimulation of the human synovial sarcoma cell line SW982 with IL-1β inhibits the expression of miR-10a, leading to the up-regulation of its target gene, T-box transcription factor 5 (TBX5), which induces the proliferation of synoviocytes and suppression of synoviocyte apoptosis in RA [143]. Thus, miR-10a is considered a biomarker of RA diagnosis and therapy [144]. 

**Osteoarthritis (OA):** OA is a severe pathological condition causing significant pain and stiffness in the joints. It is often characterized by the degradation of articular cartilage and inflammation in the joints [145]. Ma et al. have demonstrated that inflamed chondrocytes show a higher expression of miR-10a, which targets the homeobox gene HOXA1, resulting in chondrocyte apoptosis [146]. Treatment with the miR-10a antagonist is shown to reduce chondrocyte apoptosis, while agomiR-10a hastens it [146]. Moreover, silencing HOXA1 reverses the rescuing effect of the miR-10a antagonist against chondrocyte apoptosis [146]. These observations suggest that miR-10a and HOXA1 could be used as therapeutic targets against OA. Li et al. showed that higher miR-10a expression in inflamed chondrocytes leads to the down-regulation of its other target, HOXA3, which results in chondrocyte apoptosis and inhibition of their proliferation [147]. The treatment with the miR-10a inhibitor increases the survivability of the inflamed chondrocytes, but HOXA3 silencing interferes with the rescuing effect of antagomiR-10a [147]. These observations suggest that targeting miR-10a could be successfully used as a therapeutic approach to treat OA [147]. miR-10a may also serve as a negative regulator during osteoblast differentiation of human bone marrow mesenchymal stem cells and may be employed in the treatment of bone repair in osteogenic-associated diseases [148]. 

**Table 4 ijms-23-15479-t004:** The role of miR-10a in various inflammation-associated diseases.

Disease	Cell Type	Expression	Target(s)	Function	Reference(s)
Rheumatoid arthritis (RA)	FLSs	Down	IRAK4,TAK1 β-TrCP, MAP3K7	Induces inflammation	[126]
	Inflamed synoviocytes	Down	TBX5	Promotes proliferation, inhibit apoptosis	[143]
Osteoarthritis	Inflamed chondrocytes	Up	HOXA1	Induces apoptosis	[146]
	Inflamed chondrocytes	Up	HOXA3	Induces apoptosis, inhibit proliferation	[147]
Inflammatory bowel disease (IBD)	Inflamed intestinal mucosal DCs	Down	IL-12, IL-23p40, NOD2	Induces inflammation	[125]
Colitis	Intestinal epithelial and DCs	Down	IL-12, IL-23p40	Induces inflammation	[123]
Atherosclerosis	ECs	Down	HOXA1, β-TrCP, MAP3K7	Induces inflammation	[122]
Sepsis	ECs	Up	IRAK4, β-TrCP, MAP3K7	Prevents inflammation	[149]
	ECs	Up	TAK1	Prevents inflammation, inhibit vascular permeability	[127,150]
Cancer	CD34^+^ mononuclear cells of CML	Down	USF2	Promotes cell growth	[151]
	Neuroblastoma	Up	NCOR2	Promotes cell growth, induce differentiation	[128]

Abbreviations: FLSs, fibroblast-like synoviocytes; DCs, dendritic cells; ECs, endothelial cells; CML, chronic myeloid leukemia.

**Inflammatory bowel disease (IBD):** IBD is another inflammation-associated disease that involves chronic inflammation of the tissues in the digestive tract [152]. In the inflamed mucosa of IBD patients, miR-10a expression is shown to be down-regulated [125]. This was believed to be responsible for the increased expression of the target genes IL-12/IL-23p40/NOD2 and prolonged intestinal inflammation [125]. The administration of anti-TNF mAb was shown to increase miR-10a expression and down-regulate IL-12/IL-23p40/NOD2, thereby inhibiting the function of Th1 and Th17 cells to control chronic inflammation in the intestine [125]. miR-10a is also shown to be expressed in the epithelial and dendritic cells in the intestine and helps maintain intestinal homeostasis. Commensal bacteria are shown to down-regulate the expression of miR-10a in intestinal dendritic cells via TLR signaling through the MyD88 pathway [123]. The down-regulation of miR-10a is accompanied by the induction of miR-10a target genes IL-12/IL-23p40 and colitis in mice [123]. Thus, miR-10a, whose aberrant expression plays a crucial role in the IBD pathogenesis, could be used as a biomarker for the IBD. 

**Acute pancreatitis (AP):** AP, another type of inflammatory disease, is characterized by an inflamed pancreas over a short period [153]. In AP patients, miR-10a levels in the serum are found to be significantly down-regulated as compared to healthy controls and thus could be used as a biomarker for AP [124]. 

**Atherosclerosis:** Atherosclerosis, an inflammation-associated disease [154], is caused by the thickening of arteries due to plaque deposition in the inner arterial wall [155]. Several inflammatory signaling pathways promote thrombosis, which is responsible for atherosclerotic complications associated with stroke and myocardial infarction [154]. Fang et al. observed a significantly lower expression of miR-10a levels in the regions of the inner aortic arch and aortic renal branches, which are susceptible to atherosclerosis [122]. HOXA1 expression, the target of miR-10a, was found to be significantly higher in those athero-susceptible regions [122]. Moreover, the expression of two key miR-10a targets, MAP3K7 and β-TrCP, were shown to be up-regulated in miR-10a knocked-down human aortic endothelial cells and accompanied by the up-regulation of pro-inflammatory NF-κB signaling pathway and the release of pro-inflammatory cytokines, IL-6, IL-8, MCP-1, and VCAM-1 [122]. The up-regulation of MAP3K7 and β-TrCP and induction of the NF-κB pathway were observed in the athero-susceptible regions of the endothelium [122]. Overall, the above data indicate that miR-10a could be a potential biomarker for atherosclerosis and targeting miR-10a could be a potential therapeutic approach to limit inflammation associated with atherosclerosis. 

**Cancer:** Inflammation and cancer are intrinsically related. The development and progression of cancer often lead to several inflammatory responses [156]. Targeting inflammation proves to be an attractive therapeutic approach for cancer prevention [156]. A growing body of evidence indicates that miR-10a expression is de-regulated in different types of cancer. A down-regulation of miR-10a is observed in several hematological cell lines [157], CD34^+^ acute [158], and chronic [151] myeloid leukemia cells. In addition, head and neck squamous cancer cells exhibit a lower expression of miR-10a [159]. In contrast, hepatocellular carcinoma stem cells are enriched with miR-10a [129,160]. In neuroblastoma cells, retinoic acid treatment induces the expression of miR-10a, which targets nuclear receptor corepressor 2 (NCOR2) [128]. The down-regulation of NCOR2 promotes the growth and differentiation of the neuroblastoma cells [128]. In many cases, miR-10a was shown to be up-regulated in cancer cells, reflecting its role in oncogenic transformation [161,162,163,164]. Although aberrant miR-10a expression is observed in different types of cancer, it is unclear at present whether its relative expression could serve as a biomarker for cancer.

## 14. Intercellular Transfer of miR-10a via Extracellular Vesicles (EVs): Functional Implications

EVs are cell-secreted, membrane-enclosed, heterogenous bodies, which play a central role in intercellular communication [165]. miRNAs are often known to be selectively packaged into the EVs by the secreting cells [166]. The advancement of single-cell EVs and single-cell EV analysis coupled with miRNA sequencing that provides a better picture of the heterogeneity of the miRNA population in the EVs helps in understanding different phenotypic effects of distinct EV populations [167,168]. Accumulating evidence indicates that EVs actively transport miRNAs between cells, thereby influencing the phenotypes of the target recipient cells [169]. EVs-associated miRNAs often serve as biomarkers in several disease conditions, such as asthma [170], traumatic brain injury [171], respiratory diseases [172], kidney diseases [173], cardiovascular diseases [174], cancer [175], liver disease [176], and diabetic neuropathy [177]. 

Recent studies showed that EVs released from vascular endothelium (EEVs) are enriched with miR-10a, and the EEVs could deliver miR-10a to recipient cells and alter their phenotype by down-regulating the miR-10a target genes in recipient cells [127,149]. Njock et al. showed that EEVs suppress monocytic activation by inhibiting pro-inflammatory responses and up-regulating immunomodulatory responses via the transfer of miR-10a [149]. EEVs-mediated transfer of miR-10a was shown to down-regulate the expression of several genes in the NF-κB signaling pathway, such as MAP3K7, β-TrCP, and IRAK4, in recipient monocytes, and thus suppress their activation [149]. Recent studies from our laboratory indicate that vascular endothelium releases EVs into the circulation in response to a coagulation protease, factor VIIa (FVIIa), via endothelial cell protein C receptor (EPCR)-dependent activation of protease-activated receptor 1 (PAR1)-mediated cell signaling [178]. In a continuation study, Das et al. demonstrated that the expression of miR-10a is increased in endothelial cells after challenging with FVIIa [127]. FVIIa-released EEVs contain significantly higher levels of miR-10a as compared to EEVs released under basal conditions [127]. The uptake of FVIIa-EEVs but not control-EEVs by monocytes or naïve endothelial cells confers anti-inflammatory or vascular protective properties, respectively, to these cell types [127]. The incorporation of anti-miR-10a into the FVIIa-released EEVs removes the EEVs’ cytoprotective responses, indicating the crucial role of miR-10a in mediating the cytoprotective responses of FVIIa-released EEVs [127]. 

Additional studies showed that FVIIa-EEVs-mediated delivery of miR-10a to monocytes down-regulates LPS-induced pro-inflammatory responses in monocytes by targeting TAK1 and down-regulating the activation of the NF-κB signaling pathway (see Figure 6) [127]. The transfer of miR-10a by FVIIa-EEVs to endothelial cells imparted barrier protective responses against LPS by restoring the expression of the tight junction protein ZO-1 [127]. Consistent with these in vitro observations, Das et al. also showed that FVIIa administration in mice increased the levels of circulating EEVs laden with miR-10a, and these EEVs imparted cytoprotective responses in ex vivo cell model systems [127]. Alternatively, the administration of FVIIa-EEVs but not control EEVs generated from cultured murine endothelial cells protected mice against LPS-induced inflammation and barrier disruption [127]. In a further study, the same group reported that FVIIa infusion into hemophilia patients increases the level of circulating EEVs in the plasma, and these EEVs contain elevated levels of miR-10a [150]. The fusion of FVIIa-EEVs with recipient cells resulted in an anti-inflammatory phenotype in monocytes and barrier-protective responses in endothelial cells, and the transfer of miR-10a to recipient cells was responsible for these protective effects [150]. 

## 15. Atypical Functions of miR-10a

miR-10a is considered to be one of the important regulators in protein synthesis as it induces the translation of 5’-terminal oligopyrimidine (TOP) mRNAs [59]. The TOP mRNAs encode proteins such as ribosomal proteins (RPs), which regulate protein synthesis [179]. miR-10a overexpression shows a ~30% induction in protein synthesis, whereas its inhibition decreases protein synthesis by ~40% in murine embryonic stem cells [59]. The binding site of miR-10a for TOP mRNAs is located at the 5’-UTR rather than the conventional 3’-UTR position of the mRNAs, and hence is considered to be atypical [59]. Another unconventional function of miR-10a is the inhibition of HOXD4 transcription via DNA methylation [180]. miR-10a overexpression is found to increase the level of the repressive H3K27me3 epigenetic histone modification on the HOXD4 promoter, leading to the transcriptional up-regulation of the HOXD4 gene [180].

## 16. miRNAs in Immune Cell Development and Function

miRNAs are known to regulate the function of different cells in the immune system. 

**T-cells:** Mature peripheral T cells are mainly comprised helper T cells (CD4^+^ T cells), cytotoxic T cells (CD8^+^ T cells), and regulatory T cells (T_reg_ cells) [181], and miRNAs are shown to be involved in the development of functional peripheral T cell subsets. For example, miR-142 binds to the 3’-UTR of glycoprotein A repetitions predominant (GARP) mRNA, and by down-regulating GARP expression suppresses the proliferation of CD25^+^ helper T cells [182]. In helper T cells, miR-29 is known to downregulate the T helper type 1 activation in response to intracellular pathogens by targeting IFN-γ [183]. 

Numerous miRNAs, such as miR-150, miR-155, and let-7 family miRNAs, are involved in the development of effector- or central-memory cells from activated CD8^+^ T cells, driven by the function of IL-2 or IL-15 [184]. In particular, miR-150 targets K (+) channel interacting protein 1 (KChIP.1) in developing central-memory T cells, and the up-regulation of KChIP.1 determines the fate of CD8^+^ T cells to central-memory T cells [184]. 

miR-10a is known to play a pivotal role in differentiating T_reg_ cells. The expression of miR-10a is induced by transforming growth factor-β (TGF-β) or retinoic acid, which restricts the T_reg_ differentiation into follicular helper T cells via targeting transcriptional suppressor Bcl-6 and co-suppressor Ncor2 [185]. miR-146a is usually over-expressed in T_reg_ cells, but its deficiency often leads to the augmented expression of its target gene signal transducer and activator transcription 1 (Stat1), thereby resulting in the dysregulation of IFN-γ responses and breakdown of immune tolerance [186]. 

**Dendritic cells:** Several miRNAs are also known to be associated with the development and differentiation of dendritic cells (DCs). For example, miR-142 is highly expressed in DCs, and its deficiency results in the impairment of DCs’ development and their differentiation from bone marrow (BM) stem cells [187]. miR-22 acts as a negative regulator of DCs’ differentiation, probably via the down-regulation of its target gene, IRF8 (interferon regulatory factor 8) expression [188]. miR-21 and miR-34a play pivotal roles in human monocyte-derived DCs (MDDCs) differentiation, and the inhibition of miR-21 or miR-34a stalled MDDCs differentiation via the up-regulation of their targets, WNT1 and JAG1 [189]. 

**B cells:** miRNAs often modulate the development of B cells. For example, the miR-17-92 cluster, which comprises six single mature miRNAs (miR-17, -18a, -19a, -20a, -19b-1, and -92a-1) and two paralogs (miR-106a~363 and miR-106b~25) [27], plays important roles in fetal and adult BM B cell development. B cell development in miR-17-92-deficient mice is perturbed at the pro- and pre-B cell differentiation stages via the modulation of the pro-apoptotic protein Bim expression [190]. Constitutive expression of miR-34a inhibits the pro-B cell to pre-B cell transition via down-regulating the expression of Foxp1, thereby perturbing the maturation of B cells [191]. miRNAs also regulate the development of B cells in peripheral B cell maturation stages. The development of marginal zone B cells (MZB) from immature B cells in the spleen was suppressed in miR-146a deficient mice through inhibition of the Notch2 pathway that directly targets Numb [192]. miR-125b, on the other hand, perturbs B cell differentiation into plasma cells in the germinal center via targeting transcription factors IRF4 and BLIMP-1 [193].

## 17. miRNAs in Pregnancy-Related Diseases

Several pregnancy-associated diseases, such as preeclampsia (PE), HELLP (hemolysis, elevated liver enzymes, and low platelet count) syndrome, and diabetes, are often characterized by an enhanced inflammatory response, and miRNAs play vital roles in the development of these diseases. Hromadnikova et al. observed an elevated level of circulating C19MC miRNAs, such as miR-516-5p, miR-517, miR-520a, miR-525, and miR-526a, in the plasma of PE patients [194]. In another study, Hromadnikova et al. reported the up-regulation of miR-17-5p, miR-20b-5p, miR-29a-3p, and miR-126-3p in the plasma of PE patients with gestational hypertension (GH) [195]. Biro et al., by system biology tool approaches, identified miR-210 in the pathogenesis of PE [196]. The same group has observed an increase in miR-210 levels in the circulating plasma exosomes of PE patients [197]. The levels of miR-122, miR-758, and miR-133a were also shown to be up-regulated in the maternal sera of patients suffering from HELLP syndrome [198]. miR-16-5p, miR-142-3p, and miR-144-3p are also shown to be up-regulated in the sera of women with gestational diabetes mellitus (GDM) and could serve as biomarkers for GDM [199]. 

## 18. Conclusions and Future Directions

A growing body of evidence from the past two decades indicates that miRNAs not only regulate host immune responses but also modulate several inflammatory pathways. Some miRNAs are known to produce pro-inflammatory effects, whereas others impart anti-inflammatory potential. In either case, they play a major role in pathophysiological conditions. Inflammatory miRNAs drive the expression of inflammatory cytokines, which recruit the immune cells necessary for the immune clearance of pathogens. On the other hand, anti-inflammatory miRNAs mediate anti-inflammatory effectsthat are essential for the post-pathogen clearance healing process. Therefore, a balance between pro- and anti-inflammatory responses is required to maintain homeostasis in the system in which miRNAs play crucial roles. Due to their stable expression in the blood and the high efficiency of methods to measure their abundance, miRNAs can be considered a promising diagnostic tool for predicting the severity of inflammatory responses. Understanding their mechanism of action will aid in developing new therapeutic strategies to control the pathogenesis of the inflammatory disease. Targeting several miRNAs by the administration of miRNA inhibitors (anti-miRNAs) or miRNA mimics could be one of the most valuable therapeutic strategies to control miRNA-target gene expression and regulate the associated inflammatory dysregulation. The prime advantage of miRNA-based therapies as compared to other gene/protein-based approaches is that the sequences are easy to synthesize, and a few minor modifications make the sequences stable and resistant to degradation inside the biological system. Another advantage is that the multiple target specificity of a single miRNA makes them more effective in controlling disease outcomes than individual gene regulation. Special caution should be exercised in developing and using miRNA-based therapies, as too much inhibition or over-expression may lead to unintended pathological abnormalities. Despite this or any other limitations, we anticipate that the miRNA-based strategies should be highly useful in diagnosing and treating different acute and chronic inflammatory diseases. 

## Figures and Tables

**Figure 1 ijms-23-15479-f001:**
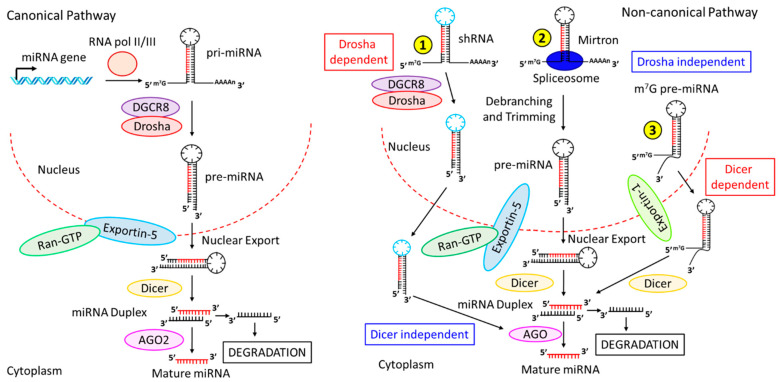
**Biogenesis pathways of miRNAs.** (**Left** panel) Canonical pathway. miRNA genes are transcribed in the nucleus by RNA Pol-II or -III to generate pri-miRNAs, which are processed by DGCR8-Drosha to produce pre-miRNAs. pre-miRNAs are transported via Exportin-5-Ran-GTP into the cytoplasm, followed by further processing by Dicer to generate the miRNA duplex. miRNA duplex then binds to AGO2, which further processes the duplex to produce one mature miRNA strand while degrading the other strand. (**Right** panel) Non-canonical pathway. The non-canonical pathway consists of multiple pathways: ① In Drosha-dependent but Dicer-independent pathway, shRNAs are processed by DGCR8-Drosha, and are readily transported from the nucleus to the cytoplasm by Exportin-5-Ran-GTP. The transported miRNA duplex is processed by AGO2 (designated as AGO in the figure) to generate the mature miRNA. ② In one Drosha-independent and Dicer-dependent pathway, Mirtrons, associated with spliceosomes, undergo debranching and trimming to generate pre-miRNAs, which are transported into the cytoplasm by Exportin-5-Ran-GTP, followed by sequential processing by Dicer and AGO1-4, respectively, to produce mature miRNAs. ③ In another Drosha-independent and Dicer-dependent pathway, (m^7^G)-capped pre-miRNAs are released into the cytoplasm via exportin-1, followed by Dicer processing to form miRNA duplex, which is further processed by AGO1-4 to generate mature miRNAs.

**Figure 2 ijms-23-15479-f002:**
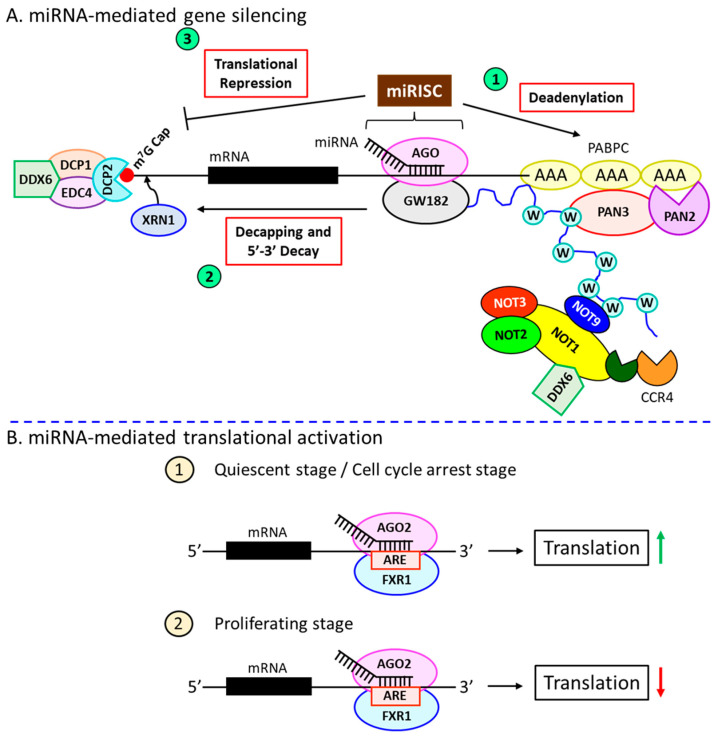
**miRNAs-mediated gene regulation.** (**A**) miRNA-mediated gene silencing mechanisms. miRNA-dependent gene silencing occurs in three independent pathways. miRNA-bound AGO forms the miRISC complex, which plays a central role in gene regulation. ① miRISC recruits the scaffolding protein, GW182, which further recruits effector complexes PAN2-3 and CCR4-NOT to induce the deadenylation process. Deadenylation is enhanced by the interaction among GW182, PABPC, and CCR4-NOT through tryptophan (W) repeats. ② GW182 recruits the decapping protein, DCP2, to the 5’-m^7^G cap of the target mRNA molecules along with other associate proteins (such as EDP4, DCP1, and DDX6), resulting in the decapping at the 5’ terminus. The decapped 5’-terminus becomes vulnerable to cleavage by XRN1 due to its 5’-3’ exonuclease activity. ③ miRISC-GW182 also interferes with the binding of ribosomes at the 5’ end of the mRNA molecules, resulting in translational repression. (**B**) miRNA-mediated translational activation. ① AGO2 and FXR1 bind to the AREs at the 3’-UTR of the target mRNAs in the quiescent stage or during cell cycle arrest and induce the translational process. ② The same binding in proliferating stage suppresses the translation.

**Figure 3 ijms-23-15479-f003:**
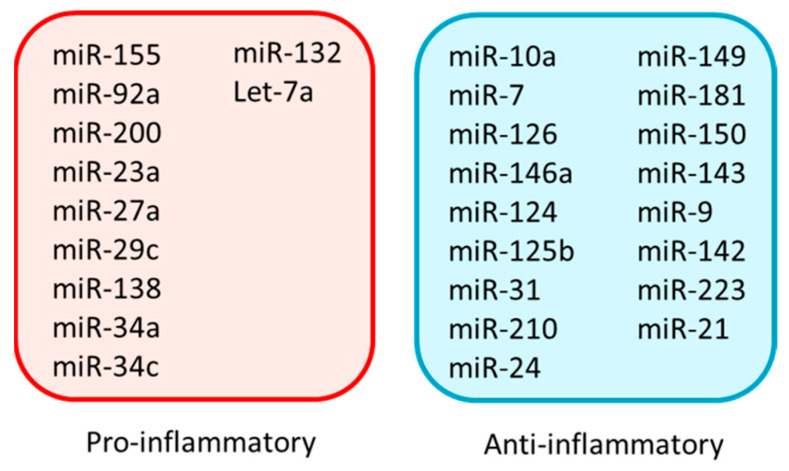
A list of pro- and anti-inflammatory miRNAs in the context of various disease conditions.

**Figure 4 ijms-23-15479-f004:**
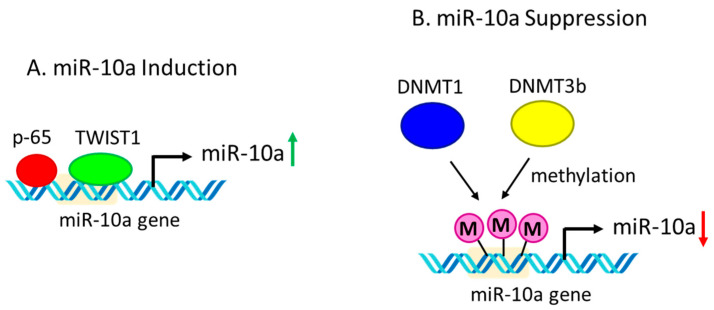
**Regulation of miR-10a expression.** (**A**) miR-10a induction. Transcription factors p65 and TWIST1 bind to the promoter region of the miR-10a gene resulting in the induction of miR-10a expression. (**B**) miR-10a suppression. DNA methyltransferases DNMT1 and DNMT3b cause promoter methylation of the miR-10a gene resulting in the down-regulation of miR-10a expression.

**Figure 5 ijms-23-15479-f005:**
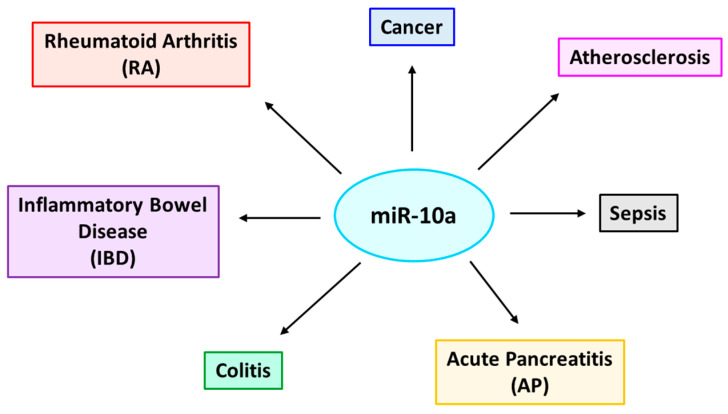
**Various inflammation-associated diseases regulated by miR-10a.** miR-10a is a key regulatory molecule that influences several inflammation-associated diseases, such as rheumatoid arthritis (RA), inflammatory bowel disease (IBD), colitis, acute pancreatitis (AP), sepsis, atherosclerosis, and cancer.

**Figure 6 ijms-23-15479-f006:**
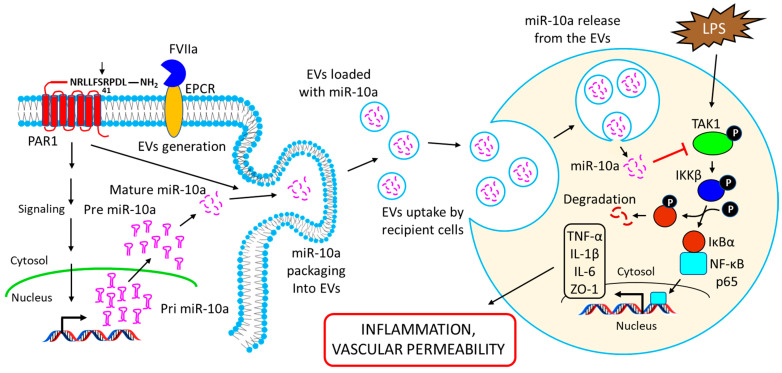
Schematic representation showing the release of miR-10a-loaded EVs from the endothelial cells and their uptake by target recipient cells to alter their phenotypic responses. FVIIa binds to EPCR on endothelial cells and activates PAR1 by proteolytic cleavage at the R41 site. The activation of PAR1 leads to the induction of miR-10a expression in the cells. PAR1 activation also triggers the release of EVs from the endothelial cells, and the FVIIa-EEVs are packaged with miR-10a. The EEVs are taken up by the target recipient cells (such as monocytes or naïve endothelial cells) via endocytosis, and the release of miR-10a from the vesicles into the recipient cell’s cytosol results in the down-regulation of the target gene’s TAK1 expression. TAK1 down-regulation impairs LPS-induced activation of the NF-κB signaling pathway and the concomitant induction of pro-inflammatory cytokines, TNF-α, IL-1β, and IL-6, as well as tight junction protein ZO-1, thereby limiting LPS-induced inflammation and vascular permeability in the recipient cells.

**Table 1 ijms-23-15479-t001:** Pro-inflammatory miRNAs and their targets and functions.

miRNAs	Cell Type	Target(s)	Functions	Reference(s)
miR-155	Macrophages	FADD, IKKɛ, Ripk1	LPS-induced miR-155 promotes inflammation by inducing TNF-α secretion	[64]
miR-92a	ECs	KLF4	Atheroprone stimuli release miR-92a-laden EVs that confer pro-inflammation to macrophages	[65]
miR-200	VSMCs	Zeb-1	miR-200 expression is increased in VSMCs from diabetic mice and induces inflammation	[66]
miR-23a	M1-macrophages	A20, JAK1, STAT6	Down-regulation of miR-23a in M1-macrophages activates NF-ĸB pro-inflammatory pathway while inhibiting anti-inflammatory pathway	[67]
miR-27a	M1-macrophages	IRF-4, PPAR-γ	Same phenotypic response as miR-23a	[67]
miR-29c	Podocytes	TTP	miR-29c up-regulation in podocytes of diabetic mice induces inflammation	[68]
miR-138	Macrophages	SIRT1	LPS stimulation induces miR-138 which activates NF-ĸB pro-inflammatory signaling pathway	[69]
miR-34a/c	Epidermal keratinocytes	LGR4	miR-34a is up-regulated in wound-edge epidermal keratinocytes of venous ulcers and induces the release of pro-inflammatory cytokines	[70]
miR-132	Primary pre-adipocytes	SIRT1	Serum deprivation induces miR-132 expression in human primary preadipocytes, which induces the release of pro-inflammatory cytokines	[71]
let-7a	ECs	IĸBβ	In atherosclerotic ECs, the increased level of let-7a activates NF-ĸB pro-inflammatory pathway	[72]

Abbreviations: ECs, endothelial cells; VSMCs, vascular smooth muscle cells.

**Table 2 ijms-23-15479-t002:** Anti-inflammatory miRNAs, their targets, and functions.

miRNAs	Cell Type	Target(s)	Functions	Reference(s)
miR-126	ECs	SPRED1, PI3KR2, VCAM-1	miR-126 exerts anti-inflammatory effects and is found to be down-regulated upon SCI	[73]
miR-146a	Macrophages	TRAF6, IRAK1	miR-146a exhibits anti-inflammatory responses and is down-regulated after the induction of diabetes	[74]
miR-124	Macrophages	STAT3, TACE	After LPS exposure, cholinergic agonists induce miR-124 expression that controls the inflammation	[75]
miR-125b	Chondrocytes	TRAF6	miR-125b expression is down-regulated in osteoarthritic conditions and the activation of NF-ĸB pro-inflammatory pathway results	[76]
miR-31	Colonic epithelial cells	GP130, IL17R, IL17RA	After CD or UC, miR-31 expression is increased to control prolonged colonic inflammation	[77]
miR-210	Chondrocytes	DR6	miR-210 expression goes down in osteoarthritic conditions causing inflammation	[78]
miR-24	Macrophages, aortic SMCs, ECs	Chi3l1	miR-24 down-regulation in AAA causing cytokines production from macrophages and SMCs, expression of cell-adhesion molecules in ECs	[79]
miR-149	Chondrocytes	TAK1	In osteoarthritis, miR-149 expression is decreased thereby activating NF-ĸB pro-inflammatory pathway	[80]
miR-181a	Kidney cells	TNF-α	In DN, miR-181a expression is down-regulated, causing the activation of TNF-α-mediated inflammatory response	[81]
miR-150	Macrophages	STAT1	Down-regulation of miR-150 by LPS activates pro-inflammatory responses	[82]
miR-143	Pulmonary epithelial cells	MyD88	Down-regulated miR-143 expression in fibromyalgia causes pro-inflammatory responses	[83]
miR-9	Macrophages	JAK1, MMP-13	oxLDL, LPS, or Alum-stimulated macrophages down-regulate miR-9 expression and activate the inflammasome	[84]
miR-142	Macrophages	IRAK-1	BCG infection down-regulates miR-142 expression in macrophages to activate NF-ĸB pro-inflammatory pathway	[85]
miR-223	HGFs	IKKα, MKP-5	Inflammatory cytokines induce miR-223 expression in HGFs which in turn induce pro-inflammatory cytokines	[86]
miR-21	Macrophages	KBTBD7	Excessive inflammation by DAMPs after MI induces miR-21 expression, which produces anti-inflammatory responses to control prolonged inflammation in post-MI	[87]

Abbreviations: ECs, endothelial cells; SCI, spinal cord injury; CD, Crohn’s disease; UC, ulcerative colitis; SMCs, smooth muscle cells; AAA, abdominal aortic aneurysm; DN, diabetic nephropathy; oxLDL, oxidized low-density lipoprotein; BCG, Bacillus Calmette–Guèrin; HGFs, human gingival fibroblasts; DAMPs, damage-associated molecular patterns; MI, myocardial infarction.

## Data Availability

Not applicable.

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
