# Peer review of "The Role of microRNAs in Inflammation"

_ijms, 2022, doi:10.3390/ijms232415479_

Round 1

Reviewer 1 Report

This review entitled "The role of small non-coding RNAs in inflammation." by Das & Rao aimed to review the advance of non-coding RNAs in inflammation. This manuscript indicated very imporant findings and written claerly. But some corrctions may be needed.  It is better add some sentences about functional relationship between extracellular vesicles and small non-coding RNAs, such as biomarkers, intercellular comminications, packaging and single cell analysis. In addition, in is better to add some sentences about small non-coding RNAs and immune cells, T cells, B cells, dendritic cells, macrophages, and Treg etc.

Reviewer 2 Report

The submitted work is interesting and nicely presented. There are informative tables and figures in the submission.

The authors summarize the findings mainly related to miRNAs. The title is about small non-coding RNAs, I miss them.

There are significant works related to pregnancy related diseases like preeclampsia, HELLP syndrome, diabetes etc. They are missing from the review, e.g. I. Hromadnikova has a significant work related them, or O. Biro, the references has to be updated and involve other significant works on this field.

I recommend to involve other small non-coding RNAs and other miRNA related works, like miRNA-7 family, which has a role indevelopment of several diseases, like cancer, pregnancy related diseases or cardiological problems.

Round 2

Reviewer 2 Report

The authors changed the title, so the content now covers the field. The changes made by the authors in the manuscript try to show this interesting topic..